# Simulators and Simulations for Extracorporeal Membrane Oxygenation: An ECMO Scoping Review

**DOI:** 10.3390/jcm12051765

**Published:** 2023-02-22

**Authors:** Wytze C. Duinmeijer, Libera Fresiello, Justyna Swol, Pau Torrella, Jordi Riera, Valentina Obreja, Mateusz Puślecki, Marek Dąbrowski, Jutta Arens, Frank R. Halfwerk

**Affiliations:** 1Engineering Organ Support Technologies, Department of Biomechanical Engineering, TechMed Centre, University of Twente, 7522 LW Enschede, The Netherlands; 2Cardiovascular and Respiratory Physiology Group, TechMed Centre, University of Twente, 7522 NH Enschede, The Netherlands; 3Institute of Clinical Physiology, National Research Council, 56124 Pisa, Italy; 4Department of Respiratory Medicine, Paracelsus Medical University Nuremberg, 90419 Nuremberg, Germany; 5Critical Care Department, Vall d’Hebron University Hospital, 08035 Barcelona, Spain; 6SODIR, Vall d’Hebron Research Institute, 08035 Barcelona, Spain; 7Cardiothoracic Intensive Care Unit, University of California, Los Angeles, CA 90095, USA; 8Barbra Streisand Women’s Heart Center, Smidt Heart Institute, Cedars-Sinai Medical Center, Los Angeles, CA 90048, USA; 9Department of Medical Rescue, Poznan University of Medical Sciences, 60-806 Poznan, Poland; 10Department of Cardiac Surgery and Transplantology, Poznan University of Medical Sciences, 61-848 Poznan, Poland; 11Polish Society of Medical Simulation, 62-400 Slupca, Poland; 12Department of Medical Education, Poznan University of Medical Science, 60-806 Poznan, Poland; 13Department of Cardio-Thoracic Surgery, Thoraxcentrum Twente, Medisch Spectrum Twente, 7500 KA Enschede, The Netherlands

**Keywords:** extracorporeal life support (ECLS), simulation training, model, classification, fidelity

## Abstract

High-volume extracorporeal membrane oxygenation (ECMO) centers generally have better outcomes than (new) low-volume ECMO centers, most likely achieved by a suitable exposure to ECMO cases. To achieve a higher level of training, simulation-based training (SBT) offers an additional option for education and extended clinical skills. SBT could also help to improve the interdisciplinary team interactions. However, the level of ECMO simulators and/or simulations (ECMO sims) techniques may vary in purpose. We present a structured and objective classification of ECMO sims based on the broad experience of users and the developer for the available ECMO sims as low-, mid-, or high-fidelity. This classification is based on overall ECMO sim fidelity, established by taking the median of the definition-based fidelity, component fidelity, and customization fidelity as determined by expert opinion. According to this new classification, only low- and mid-fidelity ECMO sims are currently available. This comparison method may be used in the future for the description of new developments in ECMO sims, making it possible for ECMO sim designers, users, and researchers to compare accordingly, and ultimately improve ECMO patient outcomes.

## 1. Introduction

Extracorporeal membrane oxygenation (ECMO) support is used for critical pulmonary and/or cardiac hemodynamic dysfunction refractory to conventional therapies [1,2]. ECMO is a type of organ support that substitutes the lungs and/or heart. To achieve this, blood is drained from a cannula located in a large vein (often femoral or jugular veins) for the peripheral configuration. Blood is oxygenated and carbon dioxide is removed. Pressurized blood flow returns back to the patient through a return cannula located in a major vein (veno-venous (VV) configuration, providing respiratory support) or in an artery (veno-arterial (VA) configuration, providing cardiorespiratory support). Gas exchange is achieved through an artificial polymethyl pentene (fiber) membrane (located in a gas interchanger or oxygenator) and the blood flow is generated by a centrifugal pump (in neonates, a roller pump may be alternatively used), see Figure 1.

Through blood contact with artificial structures and depending on the patient’s condition, the coagulation system may be activated. Therefore, patients on ECMO support usually receive anticoagulation to prevent thrombotic events. Thrombotic and hemorrhagic events, as well as infections, are the most frequent complications [4]. These complications may lead to life-threatening conditions and should promptly be identified and urgently solved. Additionally, the cannulation process (generally carried out via the Seldinger technique) is associated with a relatively high risk of complications [5]. Because of the high and life-threatening complication risks, nursing care should be performed cautiously, making in-hospital and out-hospital transports challenging. All these factors make the clinical scenario very complex for the involved clinical staff. Despite the complexity of the ECMO system and clinical environment, the use of ECMO in recent years has been increasing, especially in the adult population. Most recently, good results have been evidenced with its use during the COVID-19 pandemic and, in some instances, in refractory cardiac arrest [6,7]. This expansion of indications has increased healthcare entities’ interest in training their teams in this technique.

Results after using ECMO are typically better in high-volume centers, with the general recommendation for concentrating cases in a few centers [8,9]. These better outcomes are mainly related to a superior level of training of the ECMO team, achieved by a suitable clinical demand. However, new and low-volume centers could also reach good results when the ECMO team receives adequate multidisciplinary and high-quality training [10,11].

A simulation is an education modality that significantly benefits training medical teams working in specialized and highly technical clinical scenarios [12]. Recently, it has gained popularity for educating individuals and teams to improve patient safety and optimize healthcare quality. The Extracorporeal Life Support Organization (ELSO), the referent ECMO international society, stresses the importance of using simulation-based training (SBT) in training the ECMO team (www.elso.org accessed on 10 October 2022). Aside from the importance of debriefing-based methodologies, optimized technical equipment is essential to enhance the learning process. Various simulators designed by different companies and institutions are now available for ECMO training and are currently being used by different entities. These devices have specific features which are conceivably essential to adapt in detail to the necessities of the different entities delivering ECMO training, each with particular objectives and teaching-targeted disciplines, such as intensive care, cardiology, and clinical perfusion.

Currently, only SBT programs have been reviewed and compared, but not ECMO sims themselves. Some developers, such as Alhomsi, et al. [13], tried to compare their work to different ECMO sims but lacked guidelines for comparison, making them subjective. Therefore, this study presents a structured and objective way of classifying available ECMO sims.

## 2. Materials and Methods

Following the PRISMA (preferred reporting items for systematic reviews and meta-analyses) scoping guidelines [14], this scoping review aims to identify characteristics and specific training media to include in ECMO simulations and simulators (ECMO sims). In addition, desk research was carried out to gather relevant literature for recent developments [13,15,16,17,18,19,20,21,22,23,24,25] and commercially available ECMO sims [26,27,28,29,30,31,32,33,34,35,36,37,38,39].

### 2.1. Desk Research

The in- and exclusion criteria for the literature collection were as follows:Research material is derived from (online) databases, including: PubMed, Web of Science Core Collection, and Google (Scholar);Research material is taken from its original source;The literature contains information regarding ECMO sims;The information applies to the comparison of ECMO sims;The literature has not been retracted;No case studies are used unless the importance of the study can be argued;No review studies are included in the literature to process specific device developments;The language of the literature material should be available in English.

A list of key concepts related to ECMO sims was created, and for each key concept synonyms, narrow terms, and broader terms were drawn up and used in search queries, see Appendix A. The search results’ relevance was noted in a table to keep an overview of the processed work, e.g., the search query TITLE-ABS-KEY(ecmo AND Simulator) in Scopus Elsevier was sorted by relevance and provided 38 results of which 10 were relevant. The bibliography of all the relevant literature was read thoroughly to identify all the applicable literature. Selected literature was then reviewed by all authors and relevant information was summarized in the literature database. The examination of this literature lead to an overview of the developed and commercially available ECMO sims.

### 2.2. Fidelity Classification

To distinguish the level of fidelity between the ECMO sims, fidelity was categorized into: low-fidelity, mid-fidelity, and high-fidelity. Furthermore, the overall fidelity was derived from the median of the definition-based fidelity, component fidelity, and customization fidelity.

#### 2.2.1. Definition-Based Fidelity

To determine the fidelity classification of ECMO sims, existing definitions were used, when possible, i.e., definition-based fidelity (DBF). These definitions have been derived from established dictionaries, standards, and literature, such as the oxford dictionary, the International Organization for Standardization (ISO), and The ELSO Red Book [40] (Table A1). According to the Healthcare Simulation Dictionary [41], DBF can be divided into four sub-categories: conceptual fidelity, functional fidelity, physical fidelity, and psychological fidelity. For conceptual fidelity, the differentiation between computational, physical, or a combination of both was determined. This was deduced from the available information about the ECMO sim. The level of functional, physical, and psychological fidelity was assigned by the authors’ educated guess based on the available ECMO information. When the level of fidelity was determined to be low, a minus (−) was assigned. When mid, a plus-minus (+/−) was appointed, and a plus (+) was set to the ECMO sim when high. Eventually, the median of these last three fidelities was calculated to determine the DBF classification and was agreed upon by all co-authors. As DBF classification is subjective, this classification was weighed alongside the component fidelity and the customization fidelity for improved objectiveness.

#### 2.2.2. Component Fidelity

Component fidelity is based on the main components of ECMO: diagnostics, circuit priming, circuit monitoring, cannulation, connection ECMO/oxygenator, gas exchange, hemodynamics, weaning, decannulation, and (clinical) scenarios [42]. Based on the available information about the ECMO sim, the authors determined whether a component was included in the design or not. In case a component was included, it was specified if it was included computationally (C), physically (P), or as a combination of both. A total score out of 10 possible components determined whether the component fidelity is low, mid, or high. Low-fidelity was assigned to ECMO sims when the total ECMO components were ≤3, mid-fidelity was assigned when the total ECMO components were 4 to 7, while high fidelity was assigned when the total number of ECMO components were ≥8.

#### 2.2.3. Customization Fidelity

Customization fidelity was based on the ability to adjust ECMO sim parameters to create more diverse patient-related scenarios. Similar to the component fidelity, the authors, based on the available ECMO sim information, determined whether a parameter was included in the design. The six parameters considered for ECMO sims were sex, age, body size, race, disease and/or anatomy, and BMI/fat percentage. These parameters were chosen to influence the ECMO procedure. Race was left out of a secondary classification for computational or a combination of computational and physical ECMO sims without representation of the patient, resulting in five parameters. Low customization fidelity was based on ≤2/6 or 1/5, mid-fidelity customization was between 2/6 and 5/6 or between 1/5 and 5/5, while high fidelity customization was achieved when ≥5/6 or 5/5.

#### 2.2.4. Overall Fidelity

Finally, the median of the outcomes of all these fidelity types was calculated for each ECMO sim to determine the overall fidelity. Based on these outcomes, the ECMO sims were compared to each other and allocated to overall low-, mid-, and high-fidelity ECMO sims.

## 3. Results

Fidelity is “The ability of the simulation to reproduce the reactions, interactions, and responses of the real-world counterpart. It is not constrained to a certain type of simulation modality, and higher levels of fidelity are not required for a simulation to be successful” [41]. As stated, the different levels of fidelity do not affect the success of the simulation. However, the realism of the simulation (low-, mid-, or high-fidelity) needs to correspond to the training activity for it to be a success. For a complete list of the definitions, see Appendix B
Table A1.

### 3.1. Definition-Based Fidelity

In this review, a total of thirty ECMO sims have been found, of which 26 ECMO simulations and simulators (ECMO sims) could be processed for classification in this review, see Appendix A. The classification of the ECMO sims based on the currently existing definitions is displayed in the definition-based fidelity (DBF) table, see Appendix A.

According to our findings, 10 (38%) ECMO sims were computational (C), 7 (27%) ECMO sims were purely physical (P), and the other 9 (35%) ECMO sims were a combination of computational and physical (C + P). Furthermore, from these ECMO sims, 9 (35%) were of low-fidelity (C = 4; P = 4; C + P = 1), 12 (46%) were of mid-fidelity (C = 6; P = 1; C + P = 5), and 5 (19%) were of high-fidelity (C = 0; P = 2; C + P = 3). This shows that there are currently no high-fidelity computational ECMO sims available.

### 3.2. Component Fidelity

The results from the component fidelity were different from those results from the DBF, see Appendix A. Out of the 26 ECMO sims, 9 (35%) were of low-fidelity, and 17 (65%) were of mid-fidelity. According to component fidelity, there are no high-fidelity ECMO sims. The hemodynamic effect was mimicked in most designs as 19 ECMO sims (73%) which included this either computationally (12), physically (5), or a combination of both (2). Clinical scenarios were just as often implemented in the design (C = 10; P = 1; C + P = 8). Cannulation was second-most included in 17 (65%) of the designed ECMO sims. Contrary to cannulation, decannulation was only (specifically) included in three ECMO sims (12%) and none of the ECMO sims (0%) had included weaning to their design.

### 3.3. Customization Fidelity

The final measured fidelity is the customization fidelity, Appendix A. Most (54%) ECMO sims did not have the option to model patient-specific scenarios. This resulted in 23 (88%) ECMO sims with low-fidelity and only 3 (12%) ECMO sims with mid-fidelity. The majority (38%) of the ECMO sims were able to adjust disease and/or anatomy parameter(s) but none (0%) of the ECMO sims were able to change sex. For customization fidelity, there were no high-fidelity classified ECMO sims as well.

### 3.4. Overall Fidelity

Taking the median of the fidelity mentioned above resulted in overall fidelity, see Table 1.

Furthermore, 10 (38%) low-, 16 (62%) mid-, and no high-fidelity ECMO sims currently exist (Table 2). According to Table 2, none (0%) of the ECMO sims with a low overall fidelity managed to achieve a high definition-based, component, or customization fidelity. In fact, all ECMO sims with a low overall fidelity also have a low customization fidelity.

Table 2 also shows that 5 (31%) out of the 16 mid-fidelity ECMO sims were determined to be of high definition-based fidelity. However, none of them reached a high overall fidelity based on our method. These 16 mid-fidelity ECMO sims could also not achieve a high component or customization fidelity. 

Furthermore, Table 2 displays that only 3 (12%) out of the 26 ECMO sims could achieve a mid-fidelity level for the customization fidelity. On the other hand, 0 (0%) of the 26 ECMO sims managed to achieve high customization fidelity. Moreover, 17 (65%) out of the 26 ECMO sims achieved a mid-component fidelity status.

## 4. Discussion

This review aimed to collect current ECMO simulations and simulators (ECMO sims) and determine their level of fidelity through an objective classification method. We present a structured and objective classification method of ECMO sims created by a diverse and experienced group of users, researchers, and developers. This classification method is based on the overall ECMO sim fidelity, established by taking the median of the definition-based fidelity, component fidelity, and customization fidelity. Combined, a more objective overall fidelity was defined, resulting in a low-, mid-, or high-fidelity classification. According to this new classification, only low- and mid-fidelity ECMO sims are currently available. 

### 4.1. Fidelity Classification

ECMO sims were only incorporated in the review if it was explicitly mentioned that they could be applied for ECMO training. Only information that was provided by the developers was processed. To provide the full spectrum of correct information on all the different ECMO sims, authors or companies were asked for additional official and public documentation when necessary.

The current classification fidelity of the ECMO sims is based on existing vocabulary defined and used by experts in the field (e.g., the International Nursing Association for Clinical and Simulation Learning, INACSL [43]). Therefore, all authors were asked to apply the terminology for component classification and other fidelities to allow for unequivocal definitions in ECMO practice.

The definition-based fidelity is based on the current way of (subjectively) classifying ECMO sims. However, Alhomsi, et al. [13] compared their work to different ECMO sims but lacked guidelines for comparison, making it subjective to the assessor. Other reviews [44,45,46] often put focus on the simulation-based education aspect. Because of these reasons, an objective review study about ECMO sims does not currently exist.

Therefore, component and customization fidelity were added to the mix to objectify the classification. The latter two only require the executor to identify parameters present in the ECMO sim, making the classification for these fidelity types the primary objective. Furthermore, for this review, all authors were asked to provide feedback on the preliminary classifications to improve objectivity. Due to the different backgrounds of the authors, the authors do not only represent many of the stakeholders, but the ECMO sims could be reviewed from a broad perspective, increasing objectivity. It is therefore encouraged that future ECMO sim classifications, using this method, are performed by a diverse group of assessors.

The reviewed ECMO sims could also use this method to increase their overall fidelity. e.g., the five ECMO sims [15,31,32,38,39] with a mid-overall fidelity and a high definition-based fidelity could look into the component and customization fidelity. By including more components and/or patient-specific parameters, they could increase these fidelity levels, e.g., if Curtis Life Research [31], Erler Zimmer [32], and The Simulator Company [39] would reach a high customization fidelity, their (median) overall fidelity would change from mid to high.

### 4.2. Limitations

Despite a thorough search, our ECMO sim database may not be complete. Additionally, only articles and literature written in the English language were included, so some ECMO sims may have been left out. Multiple search engines, such as Google Scholar, Scopus, PubMed, and regular Google searches, were consulted to avert the possibility of missing out on information by the latter. All bibliographies of included literature were also screened to identify other helpful literature. Although, a careful search strategy was used, some ECMO sims had no public information and were therefore not included in this review (e.g., Fresenius Heart and Lung App). In addition, this overview is accurate until 31 October 2022. This list may be updated in the future when new ECMO sims are developed.

Not all ECMO sims include specific simulation components in the same way. For example, some included gas exchange by changing the color of the blood in the lines. In contrast, others included gas exchange by adjusting the parameters digitally, and some had a combination of both. Therefore, the tables include how certain components are processed in the ECMO sims (computational: C; physical: P; or a combination of both: C + P). However, a similar approach does not mean similar quality. This also goes for the component and customization fidelity.

A quality threshold can be established through the verification and validation of an ECMO sim. However, an ECMO sim’s fidelity level and/or goals must be allocated before a design can be verified and/or validated, because verification and validation of, e.g., a low ECMO sim focusing solely on cannulation might be completely different from a mid-fidelity whole body ECMO sim. Therefore, this was separated from our methodology and is considered to be future work. However, we emphasize the importance of the verification and/or validation of simulations and simulators in general.

### 4.3. Future Prospective

We believe this method can fill the gap for developing, comparing, and classifying ECMO sims. For new developments, the method can be used for decision making on what to include to achieve a certain level of fidelity, e.g., according to this method, high component fidelity is achieved when at least 8 out of the 10 components have been included in the design and a high customization fidelity is obtained when at least 5 out of 6 (or 5) parameters have been included. This also means that when aiming for a mid (or low) fidelity classification, a developer can estimate how many components and/or parameters to include in the design to achieve this level. For research, it is now clear what has already been developed, and what still needs improvement (e.g., customizability).

For ECMO simulation-based training (SBT), this classification method helps clinicians to select the most suitable ECMO sim for the intended training purpose, e.g., cannulation or circuit monitoring. Indeed, selecting the right ECMO sim could help to achieve the set SBT goals faster. It could therefore in the future be interesting to develop a selection tool, based on this method, to aid this decision-making process. Furthermore, objective classification of simulations and simulators, as shown in this work, could be used beyond ECMO SBT. Applying this method in other (medical) fields could help to improve development, selection, and the comparison of simulations and simulators used for SBT.

Therefore, it could be helpful if organizations, such as the Extracorporeal Life Support Organization (ELSO), would promote the standardization of the classification in an objective manner. Our method could be the foundation for this matter.

Finally, the implementation and effectiveness of virtual reality and mixed reality in an ECMO sim could be explored in the future.

## 5. Conclusions

An overall fidelity was created to objectively classify based on definition-based, com-ponent, and customization fidelity. According to our new method, no high-fidelity ECMO sims currently exist, urging for the development of a high-fidelity simulator to improve ECMO-team training and potentially improve patient out-comes. This comparison method may be used in the future for the description of newly developed ECMO sims.

## Figures and Tables

**Figure 1 jcm-12-01765-f001:**
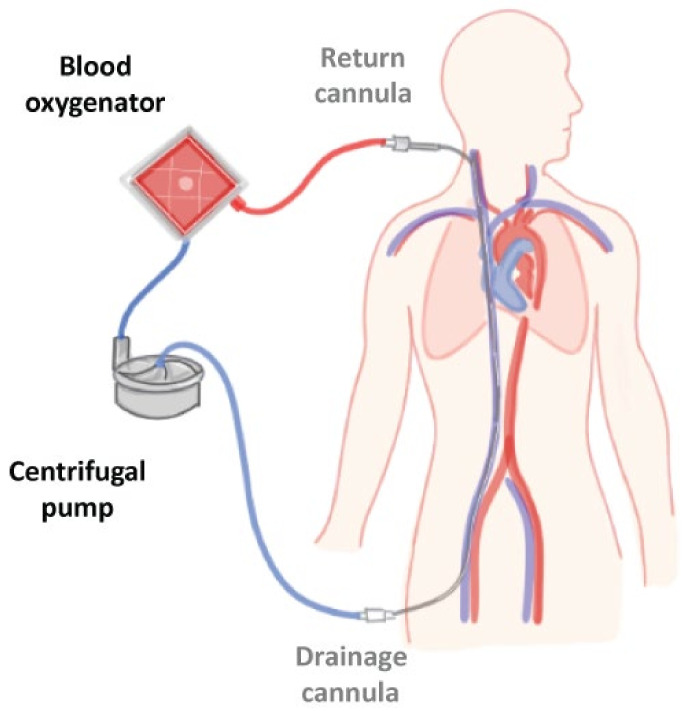
Simplified representation of an ECMO system, with a veno-venous configuration draining blood from the femoral vein, oxygenation and carbon dioxide removal, and returning blood through the jugular vein back to the patient. Adjusted image from Martins Costa, et al. [3].

**Table 1 jcm-12-01765-t001:** Overall fidelity classification of ECMO sims derived from definition-based, component, and customization fidelity.

Source	Device	Conceptual Fidelity (C, P, or C + P)	Definition-Based Fidelity (See Appendix A)	Component Fidelity (See Appendix A)	Customization Fidelity (See Appendix A)	Overall Fidelity
3-Dmed [26]	ECMO Simulation Kit	P	Low	Low	Low	**Low**
Endo, et al. [18]	Endo Circuit	P	Low	Low	Low	**Low**
Mahmoud, et al. [20]	Cannulation simulator	C + P	Low	Low	Low	**Low**
Palmer, et al. [21]	Surgical model	P	Low	Low	Low	**Low**
Thompson, et al. [24]	ECMO Initiation Simulator	P	Low	Low	Low	**Low**
Broman, et al. [16]	Aplysia	C	Low	Mid	Low	**Low**
Puslecki, et al. [23]	ECMO Therapy Simulator	C + P	Low	Mid	Low	**Low**
Telehealth Research Institute [37]	ECMOjo	C	Low	Mid	Low	**Low**
Creaplast [30]	ECMO Trainer Evolution III	C + P	Mid	Low	Low	**Low**
Palmer, et al. [22]	Percutaneous Model	P	Mid	Low	Low	**Low**
BioMed Simulations [27]	Califia Patient Simulator	C	Low	Mid	Mid	**Mid**
BioMed Simulations [28]	Califia Lung Simulator	C	Low	Mid	Mid	**Mid**
Alhomsi, et al. [13]	Modular ECMO Simulator	C + P	Mid	Mid	Low	**Mid**
Chalice [29]	Parallel Simulator	C	Mid	Mid	Low	**Mid**
Colasanti, et al. [17]	Computational ECMO Simulator	C	Mid	Mid	Low	**Mid**
Health Care Engineering Systems Center [33]	ECMO Training Simulator	C + P	Mid	Mid	Low	**Mid**
Lansdowne, et al. [19]	Orpheus Perfusion Simulator	C	Mid	Mid	Low	**Mid**
Medical Simulator [34]	Hybrids Vita	C + P	Mid	Mid	Low	**Mid**
MSE [35]	Adult ECMO Simulator	C	Mid	Mid	Low	**Mid**
PVLoops [36]	Harvi ECMO	C	Mid	Mid	Low	**Mid**
Zanella, et al. [25]	Mathematical ECMO model	C	Mid	Mid	Mid	**Mid**
Allan, et al. [15]	Integrated Skills Trainer	P	High	Low	Low	**Mid**
Texas Children’s Hospital [38]	RediStick ECMO Cannulation Trainer	P	High	Low	Low	**Mid**
Curtis Life Research [31]	Eigenflow 2 ADVANCED	C + P	High	Mid	Low	**Mid**
Erler Zimmer [32]	ECMO Trainer Professional MK2	C + P	High	Mid	Low	**Mid**
The Simulator Company [39]	E-Sim Pro	C + P	High	Mid	Low	**Mid**

**Table 2 jcm-12-01765-t002:** Summary of the overall fidelity classification of ECMO sims derived from definition-based, component, and customization fidelity.

Overall Fidelity	Definition-Based Fidelity			Component Fidelity			Customization Fidelity		
*Classification*	*Low*	*Mid*	*High*	*Low*	*Mid*	*High*	*Low*	*Mid*	*High*
Low, *n* = 10 (38%)	8 (80%)	2 (20%)	0 (0%)	7 (70%)	3 (30%)	0 (0%)	10 (100%)	0 (0%)	0 (0%)
Mid, *n* = 16 (62%)	2 (13%)	9 (56%)	5 (31%)	2 (12.5%)	14 (87.5%)	0 (0%)	13 (81%)	3 (19%)	0 (0%)
High, *n* = 0 (0%)	0 (0%)	0 (0%)	0 (0%)	0 (0%)	0 (0%)	0 (0%)	0 (0%)	0 (0%)	0 (0%)

## Data Availability

Not applicable, all data is shared in this manuscript and its supplements.

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
