# Peer review of "Simulators and Simulations for Extracorporeal Membrane Oxygenation: An ECMO Scoping Review"

_jcm, 2023, doi:10.3390/jcm12051765_

Round 1
Reviewer 1 Report
The authors conducted an interesting overview about current simulators and simulations for ECMO.
In the opinion of the reviewer, the authors might discuss dividing the tables/figures into simulators and simulations.
l. 56: Error! Reference source not found.. 5 -> Please correct.
Table 1: Are all sims commercially available, if there are differences, the authors should state this information.
Figure S6: Higher resolution is recommended.
The authors should integrate suggestions for the specific application of the various sims.
Author Response
Please see the attachment, section: "Response to Reviewer 1 Comments".

Reviewer 2 Report
The authors attempted to provide a systematic and objective approach in classifying ECMO simulators in fidelity classes. Based on their criteria for fidelity assessment, no high fidelity ECMO simulators are available. I have the following comments for the authors:
· Please provide definitions for each fidelity-component used in the review, e.g. definition based fidelity, conceptual fidelity etc. Also provide definitions for other technical terms presented in the manuscript e.g. computational or physical ECMO simulator etc. If this information is provided elsewhere in the text (e.g. tables or appendix) please make a clear referral in the text at the place of first mention of the technical term.
· Please elaborate more on the method of classifying each fidelity-component into low, mid or high. What were the used criteria and was the classification done by the authors themselves, or was it based on already published fidelity-classification information?
· Please provide the exact search-query used to retrieve the relevant literature included in this study.
· Please reformulate the text in lines 214-22, as it is unclear what is being presented.
· Discussion section: please elaborate more on the used methodology for fidelity-classification and provide clear information about the way this review helps future ECMO-simulator developers make high fidelity simulators. Have other reviews already been published in the field of ECMO simulations, and if yes how do they compare to this review? What are the key components and categories of ECMO simulators and how do they compare regarding fidelity? Please provide more information about the ECMO simulators included in this review, especially the highest-fidelity ones, and what should be done better to increase the overall fidelity of them and other ECMO simulators. The discussion section needs significant language and content editing overall.
· Other minor issues:
o Please no undefined abbreviations in the abstract, e.g sims.
o Please correct reference-source-error in line 57.
o Please correct type-error in line 192 (26 ECMO simulators?).
o Please add lines in Table 2 or correct line alignment to improve readability.
Author Response
Please see the attachment, section: "Response to Reviewer 2 Comments".

Round 2
Reviewer 2 Report
I would like to thank the authors for their efforts on addressing the issues raised during the initial review. I have the following comments/remarks:
· Were other search-terms used aside from the one provided in the example? If yes, then please provide them also. If the search-terms used to retrieve the relevant literature are too many to fit in the methods section, then they should be provided as supplementary material.
· Please reformulate the paragraph from line 315 to line 321 to improve readability.
· Please format references according to the journal's "instructions for authors".
· Some minor comments:
o Please use landscape layout to present table 1
o Please use table Option B for Table 2. Additionally, please place each number and respective percentage on the same line to improve readability. Please correct the title of table 2 by adding the word "of" (Summary of…)
o Line 343 (type error): please write "definition-based"
o Appendix 1 (type error): please use the word "cannulae" for the plural form
o Appendix 1 (type error) / Physical (simulator) definition: … for "an", offen simplified…
